# Intergenerational Transmission of Gut Microbiome from Infected and Non-Infected *Salmonella pullorum* Hens

**DOI:** 10.3390/microorganisms13030640

**Published:** 2025-03-11

**Authors:** Qing Niu, Kaixuan Yang, Zhenxiang Zhou, Qizhong Huang, Junliang Wang

**Affiliations:** 1Animal Husbandry and Veterinary Research Institute, Shanghai Academy of Agricultural Science, 2901 Beidi Road, Shanghai 201106, China; nqdwkx@163.com (Q.N.); yangkaixuan007@hotmail.com (K.Y.); shsnykxy@163.com (Z.Z.); huangqizh@163.com (Q.H.); 2Zhuanghang Comprehensive Test Station, Shanghai Academy of Agricultural Science, 888 Yezhuang Road, Shanghai 201415, China

**Keywords:** poultry, *Salmonella pullorum*, fecal microbiota transplantation, microbial structure, prediction functions

## Abstract

Pullorum disease (PD) is one of the common infectious diseases in the poultry industry in the world. Our previous study showed that gut bacterial structure has a significant difference between positive and negative hens. However, the gut bacterial basis of intergenerational transmission of PD continues to elude a scientific explanation. The present study carried out fecal microbiota transplantation (FMT) in chicks of a negative group, then fecal samples of the chicks in the control team (CT), *Salmonella pullorum* (*S. pullorum*)-negative transplantation team (PN) and *S. pullorum*-positive transplantation team (PP) were separately collected to be analyzed for microbial structure and prediction functions. Microbial diversity results revealed that there was a large difference in the gut microbiota of these three groups. *Prevotella* and *Parasutterella* with higher abundance in PN (*p* < 0.05) were transplanted from gut bacteria of *S. pullorum*-negative hens. Furthermore, the differences of the most major microbial functions (top 100) were similar in hens and chicks, including a pentose phosphate pathway and oxidative phosphorylation. The data provided a reference for exploring the intergenerational transmission and genetic mechanisms of gut microbiota associated with *S. pullorum* in poultry, as well as a theoretical basis for improving intestinal health through the rational regulation of microbiota-host interactions.

## 1. Introduction

Pullorum disease (PD), caused by infection with *Salmonella pullorum* (*S. pullorum*), is more common in young birds and transmitted both vertically and horizontally in chickens [1,2,3]. Previous studies demonstrated that PD was observed as a result of increased stocking density and in larger farms, so comprehensive and stringent strategies should be used to prevent and control this disease [4,5,6]. Although eradication programs have been carried out as a prevention and control measure, as the agglutination test results are erratic, it still causes decreased production performance and even death of poultry, as well as being a threat to public health [2,7]. Treatment of flocks with PD will not alleviate the perpetuation of the carrier state and is not recommended [8].

Fecal microbiota transplantation (FMT) refers to the transfer of the fecal microbiota from a healthy donor into the gut of a diseased recipient, which restores the composition and functionality of the intestinal microbial community and resists the colonization of pathogens [9,10]. FMT has garnered widespread attention due to its high safety, reliability, and cure rates, and has gradually become a standard clinical approach for treating diseases such as *Clostridioides* difficile infection and necrotizing colitis in both humans and animals [11]. A chicken’s gut harbors a rich and diverse microbial community; absolute counts of microbiota are around 10^10^ CFU per gram of digesta and it is populated by approximately 1000 different species [12]. These microbes form a relatively stable microecosystem with their host through mutualistic symbiosis. Collectively functioning as a superorganism, they serve as a critical biological barrier in the gut. Furthermore, the establishment and structural composition of these diverse and abundant gut microbiota provide essential support for the maturation of both the intestinal and immune systems [13]. This symbiotic relationship significantly reduces the risk of pathogenic microbial infections in the intestinal tract [14].

In our previous study, we clarified the morphological characteristics of the reproductive tracts and intestines of hens infected with *S. pullorum* and preliminarily explored the potential association between cecal microbiota and reproductive performance in hens [15]. A total of 50 genera were in greater counts in *S. pullorum*-negative hens, such as *Bacteroides*, *Desulfovibrio*, *Megamonas*, *Prevotella* and *Parasutterella*. Diminished phosphotransferase system and pentose phosphate pathway, butanoate metabolism and oxidative phosphorylation were also activated in *S. pullorum*-positive hens. However, the efficacy of most probiotics reported previously was either limited or insufficiently characterized. Further studies are needed to elucidate the role of gut microbiota in *S. pullorum* infection and its functional implications in chickens. With the advances of the gut microbiota in physiology and health, understanding the intergenerational transmission and genetic mechanism of the gut microbiome related to *S. pullorum* and developing effective interventions is paramount. The present study carried out fecal microbiota transplantation in chicks of a negative group, then fecal contents were collected to analyze the microbial structure and prediction functions at two weeks post-transplantation in the control team (CT), *S. pullorum*-negative transplantation team (PN) and *S. pullorum*-positive transplantation team (PP), respectively. This study aimed to analyze the intergenerational transmission mechanism of *S. pullorum*-associated gut microbiota in poultry and to elucidate how exogenous microbial transplantation modulates the reconstitution of intestinal microbiota and its functional dynamics in recipient hosts.

## 2. Materials and Methods

### 2.1. Ethics Statement

All procedures and the use of animals were carried out in accordance with the Guidelines for the Ethics and Animal Welfare Committee of the Shanghai Academy of Agricultural Sciences (No. SAASPZ0521041). All necessary measures were taken to ensure the chickens experienced minimal distress during this study.

### 2.2. Animal Experiment and Sample Collection

Donor New Pudong hens used in this study were obtained after three slide agglutination tests, and the results of *S. pullorum* infections texts were positive or negative all three times. The New Pudong female chicks that underwent FMT were obtained from hens confirmed to be *S. pullorum*-negative by the whole blood plate agglutination test. All animals were selected according to a unified breed standard and fed with antibiotic-free corn-soybean diets (Appendix A) from the experimental farm of the Shanghai Academy of Agricultural Sciences, Shanghai, China. Antibiotics in the feed or for any therapeutic purposes were not provided for chicks and hens after the age of three days. The hens were vaccinated according to a standard immunization protocol, whereas the chicks remained unvaccinated. Fresh feces from the 45-week-old donor hens were collected to prepare for microbiota suspension as described below. Fresh contents were separately pooled from hens with *S. pullorum*-negative and *S. pullorum*-positive, homogenized, diluted five-fold in sterile potassium phosphate buffer (0.1 M, pH 7.2) containing 15% glycerol (*v*/*v*), then immediately dispensed to cryotubes, and stored at –80 °C [16].

At first, a total of 60 3-day-old chicks were selected as recipients in this study to characterize the specific colonization. These offspring of hens negative for pullorum disease were randomly allocated to 3 groups (control team (CT), *S. pullorum*-negative transplantation team (PN) and *S. pullorum*-positive transplantation team (PP)) with 4 replicates of 5 birds each. Then these chicks were inoculated orally with 1 mL of sterile saline, feces microbial suspension from hens with *S. pullorum*-negative and *S. pullorum*-positive, respectively, once every two days for 3 days. Three and eight fecal samples from each group were randomly and individually collected in 2 mL centrifuge tubes for 16S rRNA gene sequencing at two time points: one day prior to oral administration and 14 days after oral administration. All samples were kept in an ice box for preservation and transportation and then stored at −80 °C in the laboratory [17].

### 2.3. 16S rRNA Sequencing and Bioinformatics Analysis

The gut microbiota population in the chicks of CT, PN and PP groups were analyzed using 16S rRNA gene sequencing. Microbial community genomic DNA was extracted from fecal samples using the E.Z.N.A.^®^ soil DNA Kit (Omega Bio-tek, Norcross, GA, USA) according to the manufacturer’s instructions. The DNA extract was checked on 1% agarose gel, and DNA concentration and purity were determined using a NanoDrop 2000 UV-visspectrophotometer (Thermo Scientific, Wilmington, DE, USA). Hypervariable V3–V4 region of the 16S rRNA gene with a length of approximately 468 bp was targeted for sequencing. PCR amplification was performed with gene-specific primers 338F (5′-ACTCCTACGGGAGGCAGCAG-3′) and 806R (5′-GGACTACHVGGGTWTCTAAT-3′) under the following conditions: initial denaturation at 95 °C for 3 min, followed by 27 cycles of denaturing at 95 °C for 30 s, annealing at 55 °C for 30 s and extension at 72 °C for 45 s, and single extension at 72 °C for 10 min, and end at 4 °C.

The PCR mixtures contained 5× TransStartFastPfu buffer 4 μL, 2.5 mM dNTPs 2 μL, forward primer (5 μM) 0.8 μL, reverse primer (5 μM) 0.8 μL, TransStartFastPfu DNA Polymerase 0.4 μL, template DNA 10 ng, and finally ddH_2_O up to 20 μL. PCR reactions were performed in triplicate. The PCR product was extracted from 2% agarose gel and purified using the PCR Clean-Up Kit (YuHua, Shanghai, China) according to the manufacturer’s instructions and quantified using Qubit 4.0 (Thermo Fisher Scientific, Waltham, MA, USA). Purified amplicons were pooled in equimolar and paired-end sequenced on an Illumina Nextseq2000 platform (Illumina, San Diego, CA, USA) according to the standard protocols by Majorbio Bio-Pharm Technology Co., Ltd. (Shanghai, China).

After demultiplexing, the resulting sequences were quality-filtered with fastp (0.19.6) [18] and merged with FLASH (v1.2.11) [19]. Then the high-quality sequences were de-noised using DADA2 (https://github.com/benjjneb/dada2, accessed on 15 October 2024) [18] plugin in the Qiime2 [20] (version 2020.2) pipeline with recommended parameters, which obtains single nucleotide resolution based on error profiles within samples. DADA2-denoised sequences are commonly termed amplicon sequence variants (ASVs). To minimize the effects of sequencing depth on alpha and beta diversity measure, the number of sequences from each sample was rarefied to 20,000, which still yielded an average Good’s coverage of 97.9%. Taxonomic assignment of ASVs was performed using the Naive bayes consensus taxonomy classifier implemented in Qiime2 and the SILVA 16S rRNA database (v138).

### 2.4. Statistical Analysis

Bioinformatic analysis of the gut microbiota was carried out using the Majorbio Cloud platform (https://cloud.majorbio.com, accessed on 8 February 2025). The Venn diagrams constructed using jvenn (http://jvenn.toulouse.inra.fr/app/index.html, accessed on 20 February 2025) with shared and unique ASVs and genera were used to depict the similarities and differences among the three communities [21]. The similarity among the microbial communities was determined using PCA (Principal Component Analysis) using the vegan package in R (v3.3.1). Analysis of similarities (ANOSIM) was carried out to test for significant differences among the microbial communities of different samples. A stacked bar plot conducted by R (v3.3.1) was used to identify the most abundant bacterial communities both on phylum and genus levels. Comparisons of taxonomic data at the phylum level among three groups were tested using the Kruskal–Wallis rank-sum test followed by a posthoc Dunn’s test with a Bonferroni correction. Statistical significance was accepted as *p*  < 0.05. The linear discriminant analysis (LDA) effect size (LEfSe) [22] (http://huttenhower.sph.harvard.edu/LEfSe, accessed on 18 February 2025) was performed to identify differentially abundant taxa (genus to genus) among the different samples (LDA score > 2, *p* < 0.05) [22]. The co-occurrence networks were constructed to explore the internal community relationships across the samples [23]. A correlation between two nodes was considered to be statistically robust if Spearman’s correlation coefficient was over 0.6 or less than −0.6, and the *p*-value less than 0.05. PICRUSt (Phylogenetic Investigation of Communities by Reconstruction of Unobserved States) was used to explore the functional composition based on ASV representative sequences [24]. Data were compared using a one-way analysis of variance with IBM SPSS Statistics 27.

## 3. Results

### 3.1. DNA Sequence Data of Samples Among Different Treatments

More than 2.4 million sequences were obtained from all samples, and there were 74,258 high-quality sequences per sample. The average sequence length was 423 bp. Microbial composition analysis showed that a total of 15 phylum, 24 classes, 64 orders, 107 families, 218 genera, 345 species and 1272 ASVs were identified from all samples. Good’s coverage was one for all samples.

At the phylum level, the three most dominant phyla were Firmicutes, Proteobacteria and Bacteroidota which comprised 96.2% of the total sequences in PN, 98.6% of the total sequences in PP and 97.1% of the total sequences in the CT, respectively (Figure 1A). At the genus level, the top three dominant genera were Lactobacillus, Candidatus_Arthromitus and Enterococcus belonging to the phylum Firmicutes, which comprised 81.7% of the total sequences in the PN, 70.4% of the total sequences in the PP and 87.0% of the total sequences in the CT, respectively (Figure 1B).

#### Differences in Reconstituting the Gut Microbiota Structure Between *S. pullorum*-Negative Transplantation Chicks and *S. pullorum*-Positive Transplantation Chicks

Simpson index was significantly different among the PN, PP and CT (*p* < 0.05, Appendix A). Chao, ACE and Sobs indices were significantly different between the PN and CT (*p* < 0.05, Appendix A).

Principal component analysis (PCA) at the genus level revealed a significant separation among the different groups. Principal component one (PC1) explained 53.0% of the sample variation, indicating significant differences in the gut microbiota of the three groups at two weeks post-oral administration (Figure 2A). Fecal samples collected from all three groups of chicks prior to FMT exhibited distinct clustering patterns (Figure 2A), demonstrating that there were no statistically significant differences in microbial structure among the groups prior to FMT. In contrast, transplantation of hen-derived microbiota significantly altered the gut bacterial community structure in offspring.

The VENN diagram showed the unique genera of the three groups, as well as the shared genera (Figure 2B). At the ASV level, a total of 1272 ASVs were identified from all fecal samples, with 153 of those existing in all groups defined as core ASVs (Figure 2B). The core ASVs comprised approximately 12.0% of the total ASVs while 544, 253 and 101 ASVs were uniquely identified in the PN, PP and CT at two weeks post-oral administration, respectively (Figure 2B). At the genus level, a total of 218 genera were identified from all fecal samples, with 76 of those existing in all groups defined as core genera (Figure 2C). The core genera comprised approximately 34.9% of the total genera while 58, 20 and 11 genera were uniquely identified in the PN, PP and CT at two weeks post-oral administration, respectively.

### 3.2. Identification of Pullorum Disease-Associated Bacterial Taxa Associated with Pullorum Disease That Can Be Transmitted Across Generations

At the phylum level, PN has the highest abundance of *Cyanobacteria* and *Deferribacterota* (*p* < 0.05, Figure 3A).

At the genus level, there was a significant enrichment of six genera in the PN, including *Prevotella* and *Parasutterella* (*p* < 0.05, Figure 3B). *Veillonella* has a significant enrichment in the CT (*p* < 0.05, Figure 3B). Additionally, an enrichment of one genus in the PP was found (Figure 3B). The abundances of six genera, including *Lactobacillus*, were significantly different between the PN and PP using a one-way ANOVA (*p* < 0.05, Appendix A).

### 3.3. Analysis of the Co-Occurrence Network of Microorganisms Between S. pullorum-Negative Hens and S. pullorum-Positive Hens

Analysis of the co-occurrence network of microorganisms at the genus level found that the genus of Proteobacteria is the core microbes in the PN, while the genus of Firmicutes is the core microbes in the PP and CT. The core microbes had extensive connections with microbes from other phyla (Figure 4). In addition, compared with the control group, the association between members of the Firmicutes phylum in the PP group seems to be closer (Figure 4).

### 3.4. Prediction Functions of Microbial Metabolism in the Chicks with Different Treatments

Using the KEGG pathway annotation information, a total of 335 microbial functions were obtained. We found that among the major microbial functions (top 100), the abundance of 50 microbial functions in the CT were generally weaker than that in the PN, and the abundance of 17 microbial functions in the PN were more than that in the PP and CT (*p* < 0.05, Figure 5, Appendix A), including pentose phosphate pathway, butanoate metabolism and oxidative phosphorylation.

## 4. Discussion

Pullorum is an intestinal bacterial disease caused by *S. pullorum* through both horizontal and vertical transmission. It is already known that PD can lead to serious economic losses to the poultry industry in terms of mortality, reduced growth and loss of egg production [25,26]. In the past decade, the efficacy of FMT in treating a range of intestinal infections has drawn significant attention to its potential applications [27]. It is now acknowledged that the fecal community contains a large proportion of microbial species of the large intestinal microbiota [16]. In this study, we demonstrated exogenous bacterial colonization through inter-species microbiota transplantation from hens to chicks.

In the present study, most of the genera we obtained by 16S rRNA sequencing were uncultured. High-throughput sequencing will soon enable the comprehensive sequencing of entire bacterial populations, offering a deeper understanding of evolutionary dynamics among related species [8]. Huge amounts of effective sequences were being obtained and the Good’s coverage index of each sample showed the modified sequences were comprehensive enough. The community richness is reflected in the Simpson index and the Chao, AC E and Sobs indices reflect the community richness. Simpson index was significantly different in the PN versus PP and CT, meanwhile, Chao, ACE and Sobs indices were significantly different between the PN and CT. Transplanting fecal microbes from different donors induced changes in the chick gut microbiota, including a reduction in richness and alterations in the composition. Similar studies have reached the same conclusion [15,28]. In the present study, *Firmicutes*, *Proteobacteria* and *Bacteroidota* were the preponderant phyla, the dominant genera were *Lactobacillus*, *Candidatus_Arthromitus* and *Enterococcus*. Ding et al. performed a microbiome comparison and a microbiome genome-wide association study to investigate the association among the host genetics, the gut microbiota, and PD in hens at the age of 52 weeks [29]. It investigated that the dominant phyla were *Firmicutes*, *Fusobacteria* and *Proteobacteria*; and the preponderant genera were *Lactobacillus*, *Fusobacterium*, *Peptoclostridium* and *Gallibacterium*. These divergent outcomes may result from variations in experimental protocols, particularly within PD-related studies.

LEfSe was used to further determine the taxa that most likely explain the differences among the PN, PP and CT. Differences in reconstituting the gut microbiota structure between the PN and PP, there was a significant enrichment of six genera, including *Prevotella* and *Parasutterella* in the PN. Phylogenetically divergent bacterial taxa promoted bacterial proliferation and orchestrated the succession of dominant consortia by remodeling the gut milieu and employing unique respiratory strategies alongside metal scavenging systems [30,31]. The results showed that the abundances of *Prevotella* and *Parasutterella* were significantly enriched in hens with *S. pullorum*-negative, as in our previous study [15]. These two genera might represent the microbial communities capable of achieving intergenerational transmission. *Prevotella* spp. have been isolated from various animal hosts and even occur free living in the environment. *Prevotella* may be the keystone bacteria species associated with host feed intake [32,33]. *Prevotella* spp. are proficient producers of the short-chain fatty acid propionate from arabinoxylans and fructo-oligosaccharides in vitro, and the higher the diversity of *Prevotella* spp., the more advantageous the fermenting ability of the microbiome will be for the benefit of the gut [34]. The genus of *Parasutterella* has been defined as a core component of the human and mouse gut microbiota, and has been correlated with various health outcomes [35]. Being one of the early colonizers, as well as succinate-producing commensal bacteria, *Parasutterella* may play a role in microbial interactions and infection resistance especially early in life [35]. *Mucispirillum* spp. belong to the phylum Deferribacteres and are prevalent but low abundant members of the rodent, pig and human microbiota. *Mucispirillum* schaedleri protects from *Salmonella* enterica serovar Typhimurium-induced colitis by interfering with the expression of the pathogen’s invasion machinery [36]. Although studies suggested that *Collinsella* spp. might contribute to inflammatory responses, promote systemic inflammation, and impair the intestinal mucosal barrier [37,38], Hirayama et al. illustrated that intestinal *Collinsella* may mitigate infection and exacerbation of COVID-19 by producing ursodeoxycholate [39]. In contrast, the reduction of these genera due to *S. pullorum* infection could signify their antagonistic nature towards the pathogen, suggesting their potential importance in resisting *S. pullorum* and their use in microbial intervention strategies for infection [28]. *Brevibacterium* was significantly enriched in the PP; the species are uncommon but important agents which could cause opportunistic infections in immunocompetent, as well as immunocompromised patients. However, no bacterial taxa of *Salmonella* were found in PP, this phenomenon could potentially result from an extremely low abundance of *Salmonella* in hens with *S. pullorum*-positive and highly active *Salmonella* serovar pullorum to persist in the spleen or the reproductive tract instead of the intestine in the present study [26].

PICRUSt analysis was performed to investigate the functional properties of microbiota. Using the KEGG pathway annotation information, the most important functions of the PN samples were metabolic pathways and the biosynthesis of secondary metabolites. The mechanisms of action of prebiotics and probiotics come through the production of organic acids, activation of the host immune system, and production of antimicrobial agents. Many probiotic preparations contain high numbers of lactobacilli that normally produce large quantities of volatile fatty acids such as formic acid. The incorporation of these into feed has been shown to inhibit gut colonization by zoonotic serovars of *Salmonella* [8,40]. The probiotics mentioned in prior studies were primarily cultured strains; however, their efficacy remains poorly characterized, and the majority of existing research is observational, lacking mechanistic insights.

The use of competitive exclusion gut flora preparations has a protective effect on the normal flora in animal intestines. There is the possibility of a very intimate interaction between host bacteria and pathogen in the intestine [7], one area for future *Salmonella* control exploration is the development of probiotic organisms which have a rational basis for protection. The data obtained in this study could be utilized to optimize both surveillance systems and biological interventions targeting intestinal carriage mechanisms.

## 5. Conclusions

In conclusion, FMT implementation altered the gut microbial composition, resulting in variations in microbial metabolic pathways and functions, with *Prevotella* and *Parasutterella* potentially representing microbial communities capable of achieving intergenerational transmission from PN. The data provided a reference for intergenerational transmission mechanism of *S. pullorum*-associated gut microbiota in poultry and delineated the ecological principles through which exogenous microbiome transplantation orchestrates structural and functional reassembly of recipient intestinal ecosystems.

## Figures and Tables

**Figure 1 microorganisms-13-00640-f001:**
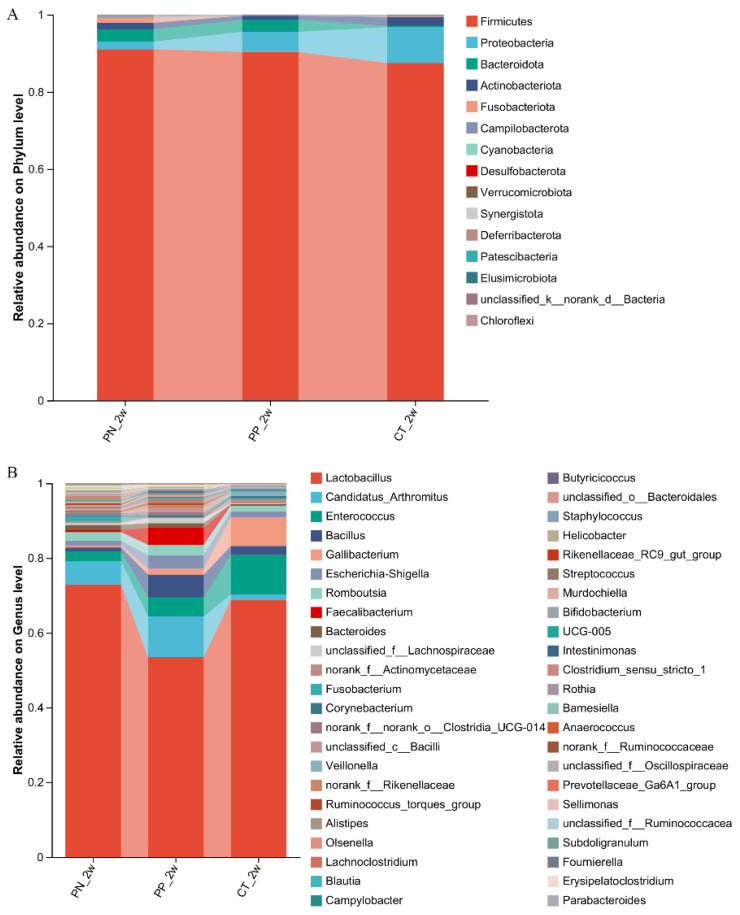
(**A**,**B**) Community barplot analysis. (**A**) was at phylum, (**B**) was at genus level. PN: *S. pullorum*-negative transplantation team, PP: *S. pullorum*-positive transplantation team, CT: control team, 2w: at two weeks post-oral administration.

**Figure 2 microorganisms-13-00640-f002:**
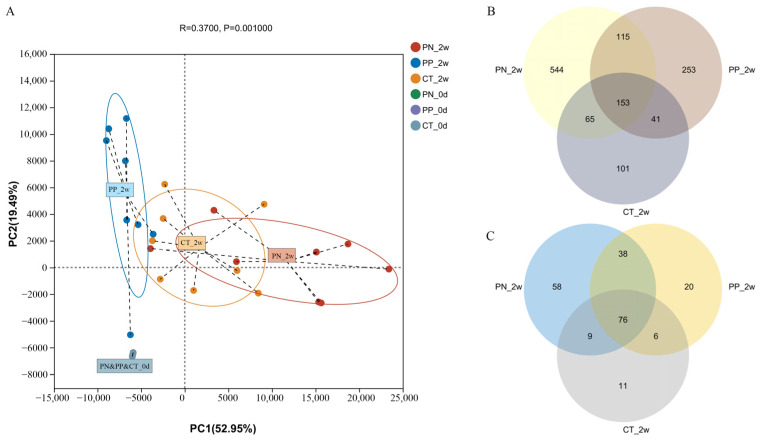
(**A**–**C**) Principal component analysis (PCA) analysis (**A**) and Venn diagrams at ASV (**B**) and genus (**C**) levels among the PN, PP and CT. PN: *S. pullorum*-negative transplantation team, PP: *S. pullorum*-positive transplantation team, CT: control team, 2w: at two weeks post-oral administration, 0d: the last day before carried out FMT.

**Figure 3 microorganisms-13-00640-f003:**
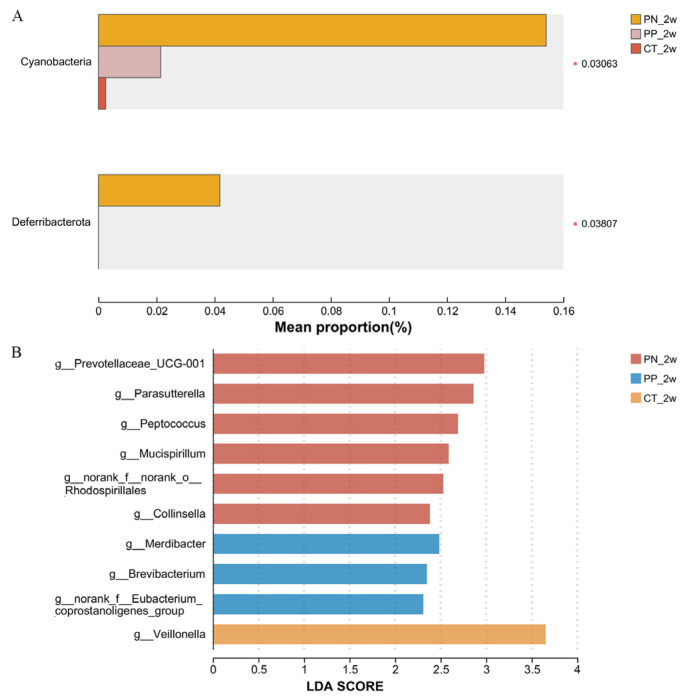
(**A**,**B**) Identification of differential microbes. (**A**) was comparisons of taxonomic data at phylum, (**B**) was Linear discriminant analysis (LDA) effect size (LEfSe) analysis at the genera level. *, the significance level of the mean difference was 0.05. PN: *S. pullorum*-negative transplantation team, PP: *S. pullorum*-positive transplantation team, CT: control team, 2w: at two weeks post-oral administration.

**Figure 4 microorganisms-13-00640-f004:**
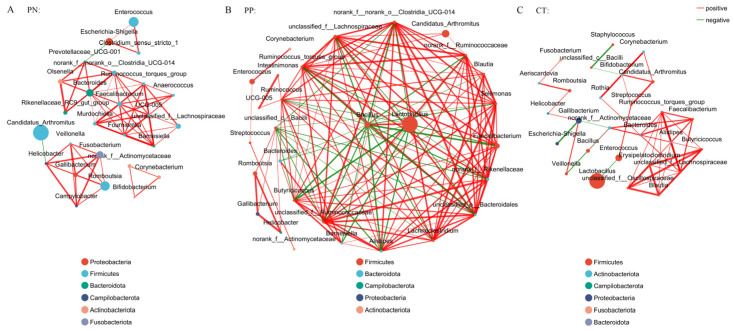
(**A**–**C**) Microbial co-occurrence network analysis at the genus level. Node size indicates relative abundance. Red edges indicate a significant positive correlation (*p* < 0.05), and green edges indicate a significant negative correlation (*p* < 0.05). PN: *S. pullorum*-negative transplantation team, PP: *S. pullorum*-positive transplantation team, CT: control team, 2w: at two weeks post-oral administration.

**Figure 5 microorganisms-13-00640-f005:**
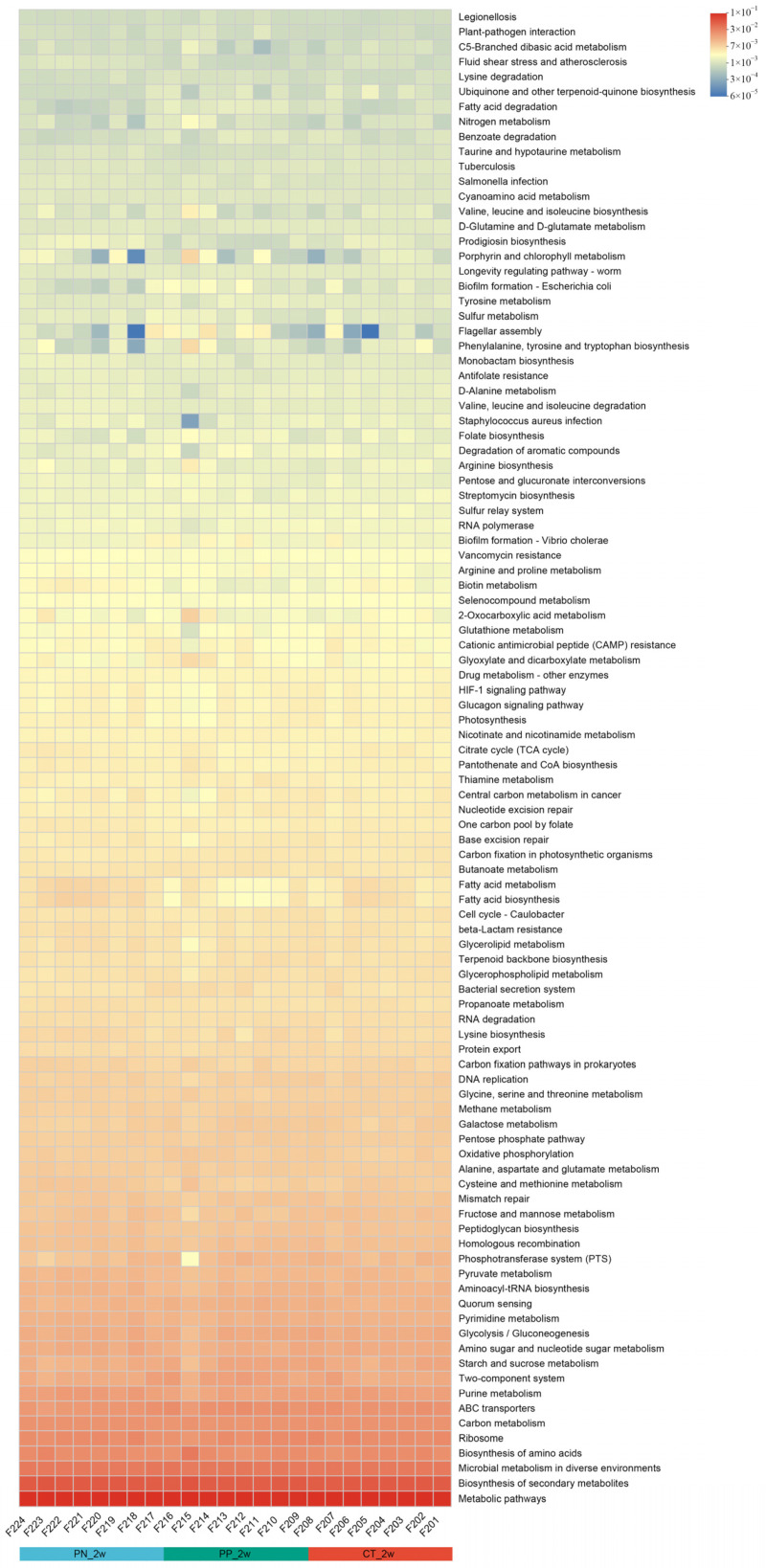
Microbial functional composition. Heatmap of the functional composition (top 100) of microbiota. PN: *S. pullorum*-negative transplantation team, PP: *S. pullorum*-positive transplantation team, CT: control team, 2w: at two weeks post-oral administration.

## Data Availability

All obtained raw sequence datasets have been deposited into the National Center for Biotechnology Information (NCBI) Sequence Read Archive (SRA) with the accession number PRJNA1225408.

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
