# Peer review of "Intergenerational Transmission of Gut Microbiome from Infected and Non-Infected Salmonella pullorum Hens"

_microorganisms, 2025, doi:10.3390/microorganisms13030640_

Round 1
Reviewer 1 Report
Comments and Suggestions for Authors
Intergenerational Transmission of Gut Microbiome from Infected and Non-infected Salmonella Pullorum Hens
This study aimed to analyze intergenerational transmission and mechanisms of gut microbiota associated with S. pullorum in poultry, as well as a theoretical basis for improving intestinal health through the rational regulation of microbiota-host interactions. The results revealed that FMT implementation altered the gut microbial composition, leading to variations in microbial metabolic pathways and functions.
The abstract is long.
L45-50: remove, irrelevant
L80-81: remove. More clear objectives are expected.
L90 (S1): more details are required about the diet and the chemical composition.
L98: more details are required about the chicks, age of hens, vaccinations, etc.
L249-250: repetition
Author Response
Thank you for your valuable comments and kind consideration of revision. The manuscript has been revised considering all the comments. Any changes made based on your comments are highlighted with blue color fonts. The point-by-point responses to your comments are listed below.
Comments 1: The abstract is long.
Response 1: Thank you so much for the important remind. The abstract has been streamlined with 198 words. L11-27
Comments 2: L45-50: remove, irrelevant
Response 2: Thank you very much for your comments. “As early as the 4th century AD, Ge Hong, a Chinese physician recommended fecal suspensions for food poisoning and severe diarrhea, and influential Chinese physician Li Shizhen recommends a similar 'yellow soup' for abdominal complaints 1,200 years later [11]. In 1958, US surgeon Ben Eiseman and colleagues treat four cases of pseudo-membranous enterocolitis using fecal enemas, marking the formal documentation of FMT technology [10].” has been deleted.
Comments 3: L80-81: remove. More clear objectives are expected.
Response 3: Thank you for your professional comment. It has been revised. “This study aimed to analyze the intergenerational transmission mechanism of S. pullorum-associated gut microbiota in poultry and to elucidate how exogenous microbial transplantation modulates the reconstitution of intestinal microbiota and its functional dynamics in recipient hosts.” L74-77
“The results revealed that FMT implementation altered the gut microbial composition, leading to variations in microbial metabolic pathways and functions.” has been removed.
Comments 4: L90 (S1): more details are required about the diet and the chemical composition.
Table S1 Composition of experimental chicken diet.
|
Diet composition |
Content (%) |
|
Corn |
65 |
|
Soybean meal |
29 |
|
Fish meal |
1 |
|
Premix1 |
5 |
|
Diet chemical composition |
Content |
|
Crude protein (%) |
18.78 |
|
Metabolizable energy (Mcal/kg) |
2.75 |
|
Crude fiber (%) |
2.91 |
|
Ether extract (%) |
3.90 |
|
Calcium (%) |
0.91 |
|
Total phosphorus (%) |
0.65 |
|
Sodium chloride |
0.42 |
|
Lysine (%) |
0.98 |
|
Methionine (%) |
0.32 |
|
Cystine |
0.33 |
|
Threonine |
0.73 |
|
Tryptophan |
0.22 |
1Premix: VA (IUkg): 165000; VD3 (IU/kg): 60000; VE (IU/kg): 600; VK3 (mg/kg): 60; VBI (mg/kg): 40; VB2 (mg/kg): 130, VB6 (mg/kg): 70; VB12 (mg/kg): 0.375; Niacin (mg/kg): 650; Pantothenic acid (mg/kg): 250; Folic acid (mg/kg): 22.5; Biotin (mg/kg): 1.75; Iron (mg/kg): 1300; Copper (mg/kg): 220; Manganese (mg/kg): 2450; Zinc (mg/kg):2200; Iodine (mg/kg):20; Selenium (mg/kg): 6.
Response 4: Thank you very much for your kindly reminding. The diet chemical composition has been added.
Comments 5: L98: more details are required about the chicks, age of hens, vaccinations, etc.
Response 5: Thank you very much for your comment, more details about the chicks (breed, sex, age, vaccinations and group information) and hens (breed, age, vaccinations) have been added. L85-104
Comments 6: L249-250: repetition
Response 6: Thank you for pointing this out, I agree with this comment. Therefore, we have revised it to “In this study, we demonstrated exogenous bacterial colonization through inter-species microbiota transplantation from hens to chicks.” L258-260

Reviewer 2 Report
Comments and Suggestions for Authors
The manuscript aimed to analyze the intergenerational transmission and mechanisms of intestinal microbiota associated with S. pullorum in poultry. Although the manuscript is interesting, it is necessary that the objective be very similar in both the abstract and the introduction since they differ a little. Furthermore, the description of the treatments is not entirely clear, and the authors should try to make it more understandable. Some more specific points are shown in the attached file

Author Response
Comments 1: The manuscript aimed to analyze the intergenerational transmission and mechanisms of intestinal microbiota associated with S. pullorum in poultry. Although the manuscript is interesting, it is necessary that the objective be very similar in both the abstract and the introduction since they differ a little. Furthermore, the description of the treatments is not entirely clear, and the authors should try to make it more understandable. Some more specific points are shown in the attached file.
Response 1: Thank you very much for your valuable comments and kind consideration of revision. The manuscript has been revised considering all the comments. Any changes made based on your comments are highlighted with red color fonts. The point-by-point responses to your comments are listed below.
Comments 2: L19-20: Check the sentence as it seems it is not complete.
Response 2: Thank you for your careful check. “there was” has been added. L19
Comments 3: L24: Change “Microbial” to “microbial ”
Response 3: Thank you for your thorough review. As the abstract was long, it has been streamlined within 200 words. Similar errors have been corrected. L320
Comments 4: L70: Change “their” to “its”
Response 4: Thank you very much for your comment, it has been revised. L66
Comments 5: L80-82: Delete, introductions usually end with just the objective and never present results.
Response 5: Thank you for pointing this out, I agree with this comment and it has been deleted. L77
Comments 6: L92-93: The chicks that underwent the fecal microbiota transplant were the chicks
Response 6: Thank you for your professional comment. It has been revised. L87-88
Comments 7: L94: delete “selected”.
Response 7: Thank you for your careful check and it has been deleted. L89
Comments 8: L109: Change “other day” to “two days”
Response 8: Thank you for your professional comment. It has been revised. L106
Comments 9: L119: Change “with” to “using a”
Response 9: Thank you for your professional comment. It has been revised. L116
Comments 10: L127: “contained”
Response 10: Thank you for your careful check and it has been revised. L124
Comments 11: L129: Check how “ddH2O” should be placed
Response 11: Thank you for carefully checking the details. and it has been revised. L126
Comments 12: L156-157: A nonparametric test with parametric interval tests? ls it possible to combine analyses? Perhaps the appropriate test
Response 12: Thank you for your professional comment. Comparisons of taxonomic data at phylum level among three groups had been tested using the Kruskal-Wallis rank-sum test followed by posthoc Dunn's test with a Bonferroni correction, and it showed the same results. L154-155
Comments 13: L179: “70.42%” Standardize the number of significant figures
Response 13: Thank you so much for your careful check and the number of significant figures meets the standard. L178, 190, 205 and 209
Comments 14: L191: Include that this is explained by principal component 1 (PC1)
Response 14: Thank you for your professional comment. It has been changed to “Principal component 1 (PC1) explained 53.0% of the sample variation, indicating significant differences in the gut microbiota of the three groups at 2 weeks post-oral administration”. L190-192
Comments 15: L201: “three groups, as well”
Response 15: Thank you very much for your comment, it has been revised. L202
Comments 16: L221: PP or CT, please check
Response 16: Thank you so much for your careful check, it has been revised. L223
Comments 17: L257 and 268: In the present study, ...
Response 17: Thank you very much for your comment, it has been revised. L261 and 272
Comments 18: L271: Proteobacteria; and the preponderant
Response 18: Thank you very much for your comment, it has been revised. L278
Comments 19: L304: immunocompetent, as well as
Response 19: Thank you very much for your comment, it has been revised. L313
Comments 20: L323-324: Check the writing of this sentence
Response 20: Thank you for your professional comment. It has been changed to “The data obtained in this study could be utilized to optimize both surveillance systems and biological interventions targeting intestinal carriage mechanisms”. L333-335
Comments 21: L323-324: that there is a large, L329: hens, and similar
Response 21: Thank you very much for your professional comments. Other reviewer think that “Transplanting hen fecal microbiota altered the gut bacterial community structure of the offspring. Microbial diversity results revealed that a large difference in the gut microbiota of these three groups. Prevotella and Parasutterella with higher abundance in PN were transplanted from gut bacteria of S. Pullorum-negative hens, similar results were obtained in our previous study. Furthermore, the differences of the most major microbial functions (top 100) were similar in hens and chicks, including Metabolic pathways, Biosynthesis of secondary metabolites, Microbial metabolism in diverse environments, pentose phosphate pathway and oxidative phosphorylation.” is not necessar (this part is partilly repeated in the next lines). So it has been deleted. L337

Reviewer 3 Report
Comments and Suggestions for Authors
This article provides information on the intergenerational transmission of gut microbiome from infected and non-infected Salmonella Pullorum hens. It is in general appropriately organized, carried out and written, however there are some points that should be corrected or clarified.
L35-36: "...PD was observed as a result of increased stocking density and in larger farms; so comprehensive..."
L47: "recommended"
L48: "treated"
L53: "...community; absolute..."
L55-59: Too large and confounding sentence. Please split and rephrase
L62: "...study, we clarified..."
L65: "A total of 50 genera were in greater counts in S. Pullorum-negative hens..."
L68: "...were also activated in S. Pullorum-positive hens."
L68-70: Please rephrase and use scientific language
L74-75: Please rephrase "were collected to be analyzed microbial structure"
L80-82: This sentence is not necessary here. Please delete
L94: Please delete the second "selected"
L109: Three and eight?
L116: Please delete "respectively"
L140-141: Please rephrase
L172: "...that a total..."
L193-195: Please rephrase. What do you mean?
L227: "...in PN, while, the genus..."
L249: "As already known, PD can lead..."
L258-260: Please rephrase
L264: "...different in PN versus PP and CT..."
L268: "In the present study..."
L270-272: When? Could you please provide details of the study?
L272: "might be leaded"?
L281: "...S-Pullorum negative as in our previous study [15]."
L317-318: Please rephrase
L326-333: I think that it is not necessar (this part it is partilly repeated in the next lines)
Comments on the Quality of English LanguageThe English could be improved to more clearly express the research
Author Response
Comments 1: This article provides information on the intergenerational transmission of gut microbiome from infected and non-infected Salmonella Pullorum hens. It is in general appropriately organized, carried out and written, however there are some points that should be corrected or clarified.
Response 1: Thank you for your valuable comments and kind consideration of revision. The manuscript has been revised considering all the comments. Any changes made based on your comments are highlighted with green color fonts. The point-by-point responses to your comments are listed below.
Comments 2: L35-36: "...PD was observed as a result of increased stocking density and in larger farms; so comprehensive..."
Response 2: Thank you for pointing this out, it has been revised. L34-35
Comments 3: L47: "recommended", L48: "treated"
Response 3: Thank you very much for your comments. Other reviewer think that “As early as the 4th century AD, Ge Hong, a Chinese physician recommended fecal suspensions for food poisoning and severe diarrhea, and influential Chinese physician Li Shizhen recommends a similar 'yellow soup' for abdominal complaints 1,200 years later [11]. In 1958, US surgeon Ben Eiseman and colleagues treat four cases of pseudo-membranous enterocolitis using fecal enemas, marking the formal documentation of FMT technology [10].” is irrelevant and it has been deleted.
Comments 4: L53: "...community; absolute..."
Response 4: Thank you very much for your comment, it has been revised. L48
Comments 5: L55-59: Too large and confounding sentence. Please split and rephrase
Response 5: Thank you very much for your professional comment, it has been split and rephrased. L50-55
Comments 6: L62: "...study, we clarified..."
Response 6: Thank you very much for your comment, it has been revised. L57
Comments 7: L65: "A total of 50 genera were in greater counts in S. Pullorum-negative hens..."
Response 7: Thank you very much for your comment, it has been revised. L60
Comments 8: L68: "...were also activated in S. Pullorum-positive hens."
Response 8: Thank you very much for your comment, it has been revised. L63-64
Comments 9: L68-70: Please rephrase and use scientific language
Response 9: Thank you very much for your professional comment, it has been revised. L64-67
Comments 10: L74-75: Please rephrase "were collected to be analyzed microbial structure"
Response 10: Thank you very much for your professional comment, it has been revised. L71
Comments 11: L80-82: This sentence is not necessary here. Please delete
Response 11: Thank you for pointing this out, I agree with this comment and it has been deleted. L77
Comments 12: L94: Please delete the second "selected"
Response 12: Thank you for your careful check and it has been deleted. L89
Comments 13: L109: Three and eight?
Response 13: “Three and eight fecal samples from each group were randomly and individually collected in 2 ml centrifuge tubes for 16S rRNA gene sequencing at two time points: one day prior to oral administration and 14 days after oral administration”. L106-109
Comments 14: L116: Please delete "respectively"
Response 14: Thank you for your professional comment and it has been deleted. L113
Comments 15: L140-141: Please rephrase
Response 15: Thank you for your professional comment. It has been changed to “DADA2-denoised sequences are commonly termed amplicon sequence variants (ASVs)” . L137-138
Comments 16: L172: "...that a total..."
Response 16: Thank you for your professional comment and it has been revised. L170
Comments 17: L193-195: Please rephrase. What do you mean?
Response 17: Thank you for your professional comment. It has been changed to “Fecal samples collected from all three groups of chicks prior to FMT exhibited distinct clustering patterns (Figure 2A), demonstrating that there were no statistically significant differences in microbial structure among the groups prior to FMT. In contrast, transplantation of hen-derived microbiota significantly altered the gut bacterial community structure in offspring”. L192-196
Comments 18: L227: "...in PN, while, the genus..."
Response 18: Thank you for your professional comment and it has been revised. L229
Comments 19: L249: "As already known, PD can lead..."
Response 19: Thank you for your professional comment and it has been revised. L253
Comments 20: L258-260: Please rephrase
Response 20: Thank you for your professional comment and it has been revised to “High-throughput sequencing will soon enable the comprehensive sequencing of entire bacterial populations, offering a deeper understanding of evolutionary dynamics among related species”. L262-264
Comments 21: L264: "...different in PN versus PP and CT..."
Response 21: Thank you for your professional comment and it has been revised. L268
Comments 22: L268: "In the present study..."
Response 22: Thank you for your professional comment and it has been revised. L272
Comments 23: L270-272: When? Could you please provide details of the study?
Response 23: Thank you for your professional comment. Some details have been added. L274-277
Comments 24: L272: "might be leaded"?
Response 24: Thank you for your professional comment and it has been revised. L279-280
Comments 25: L281: "...S-Pullorum negative as in our previous study [15]."
Response 25: Thank you for your professional comment and it has been revised. L289
Comments 26: L317-318: Please rephrase
Response 26: Thank you for your professional comment and it has been revised. L326-328
Comments 27: L326-333: I think that it is not necessar (this part it is partilly repeated in the next lines)
Response 27: Thank you for your professional comment and it has been deleted.. L337

Round 2
Reviewer 2 Report
Comments and Suggestions for Authors
The quality of the manuscript increased significantly after the first review. In addition, the authors answered each of the questions asked. In this sense, the manuscript has the necessary quality for publication.